# Relationship between dental experiences, oral hygiene education and self-reported oral hygiene behaviour

Maxi Mueller[1]*, Sarah Schorle[2], Kirstin Vach[3], Armin Hartmann[2], Almut Zeeck[2], Nadine Schlueter[1]

1 Faculty of Medicine, Division for Cariology, Department of Operative Dentistry and Periodontology, Medical Center—University of Freiburg, Freiburg, Germany, 2 Faculty of Medicine, Department of Psychosomatic Medicine and Psychotherapy, University of Freiburg, Freiburg, Germany, 3 Faculty of Medicine, Institute of Medical Biometry and Statistics, Medical Center—University of Freiburg, Freiburg, Germany

* maxi.mueller@uniklinik-freiburg.de

**Data Availability Statement:** All relevant data are within the paper and its Supporting Information files.

## Abstract

Many preventive approaches in dentistry aim to improve oral health through behavioural instruction or intervention concerning oral health behaviour. However, it is still unknown which factors have the highest impact on oral health behaviours, such as toothbrushing or regular dental check-ups. Various external and internal individual factors such as education, experience with dentists or influence by parents could be relevant. Therefore, the present observational study investigated the influence of these factors on self-reported oral heath behaviour. One hundred and seventy participants completed standardized questionnaires about dental anxiety (Dental Anxiety Scale (DAS), and dental self-efficacy perceptions (dSEP)). They also answered newly composed questionnaires on oral hygiene behaviours and attitudes, current and childhood dental experiences as well as parental oral hygiene education and care. Four independent factors, namely attitude towards oral hygiene, attitude towards one's teeth, sense of care and self-inspection of one's teeth were extracted from these questionnaires by rotating factor analysis. The results of the questionnaires were correlated by means of linear regressions. Dental anxiety was related to current negative emotions when visiting a dentist and negative dental-related experiences during childhood. High DAS scores, infantile and current negative experiences showed significant negative correlations with the attitude towards oral hygiene and one's teeth. Dental anxiety and current negative dental experiences reduced participants' dental self-efficacy perceptions as well as the self-inspection of one's teeth. While parental care positively influenced the attitude towards one's teeth, dental self-efficacy perceptions significantly correlated with attitude towards oral hygiene, self-inspection of one's teeth and parental care. Dental anxiety, dental experiences, parents' care for their children's oral hygiene and dental self-efficacy perceptions influence the attitude towards oral hygiene and one's own oral cavity as well as the autonomous control of one's own dental health. Therefore, oral hygiene instruction and the development of patient-centred preventive approaches should consider these factors.

**Funding:** The study was funded by the Medical Faculty of the University of Freiburg. The article processing charge was funded by the Baden-Wuerttemberg Ministry of Science, Research and Art and the University of Freiburg in the funding programme Open Access Publishing.

**Competing interests:** The authors have declared that no competing interests exist.

## Introduction

Over the last decades, prevention of oral diseases came more and more in focus as there is evidence that oral health is not only a matter of oral wellbeing and quality of life; it can also affect the overall health. Therefore, it is important not only to maintain the chewing function of the teeth, but also to decrease inflammatory processes or pain and maintain the aesthetics for social and psychological reasons.

Oral health behaviours summarise a wide range of measures designed to help preventing any alteration of oral health. In addition to the passive avoidance of harmful behaviour, such as smoking or the consumption of tooth-damaging foods, this also includes behavioural patterns that must be actively practised. Especially for active measures, their relevance must be conveyed to the patient and their implementation must be internalised so that they are maintained over the course of the patient's life. The most widespread and individually enforceable measures are oral hygiene measures such as regular and careful tooth brushing with a fluoride toothpaste and interdental cleaning, as well as dental check-ups and the implementation of any necessary therapies.

The factors determining and influencing individual oral health behaviour and oral hygiene habits are largely unknown and the effectiveness of professional oral hygiene instruction and education vary widely [1,2]. Further influences on the formation of oral hygiene behaviour, for example individual psychological variables, such as one's dental self-efficacy perception and dental anxiety, parental oral hygiene education, dental experiences in child- and adulthood, have been discussed to offer explanatory approaches [3–6].

Various studies have shown that oral hygiene behaviour, like toothbrushing frequency, interdental space hygiene, dental visits, the use of fluoride-containing oral hygiene products and the reduction of sugar intake were correlated with personality traits and psychological variables, such as conscientiousness, neuroticism, extraversion, tolerance or optimism, and other psychological variables, like self-efficacy, self-confidence and appreciation of one's own body [5,7–13]. The studies indicate, although very differently and partly contradictory, that individual personality traits can influence oral hygiene behaviour and could be of possible use in changing oral hygiene behaviours [14–16].

Apart from individual psychological variables, extrinsic factors, like anxiety-provoking negative dental experiences, dental anxiety as well as parental and social influences, should be considered. Negative experiences with dentists in adulthood and especially in the vulnerable phase of childhood can cause dental anxiety [17,18]. Dental anxiety can later lead to avoiding visits to the dentist and affect oral health and oral health behaviour [19,20]. Parental oral health behaviours, including frequency of toothbrushing and dental visits, dental anxiety, self-perception of one's own oral hygiene, caries experience, attitude towards oral hygiene, oral hygiene related self-efficacy perception and maternal care have an effect on children's frequency of toothbrushing and dental visits and dental anxiety [21–24]. This shows that a large and diverse spectrum of possible factors might be relevant for oral health behaviour. As mentioned, single factors have already been investigated regarding their role, however, the interaction of them have never been in the focus of research.

Therefore, the aim of the study was to explore whether and to what extent individual personality variables and experiences affect oral health behaviours. Here, a part of this study is presented. The objectives of this study part were correlations between parameters of dental experiences, parental care and oral hygiene education with self-reported oral hygiene behaviours, such as toothbrushing frequency, the use of interdental hygiene aids, frequency and intention of dental visits and attitudes towards those. The null-hypotheses were (1) that there

is no relationship between dental experiences, parental care and oral hygiene education and (2) that these parameters do not determine the self-reported behaviour.

## Materials and methods

### Study design and setting

The manuscript presents data from a larger interdisciplinary prospective non-disguised video observation study analysing brushing habits [25] and including a clinical assessment of plaque quantity (PI; [26]) as a parameter for effective brushing performance and a questionnaire-based survey on 170 healthy volunteers. The methodology was based on a previous study and did not include any interventions [25]. The questionnaire-based survey included socio-demographic data, self-report of oral hygiene attitudes and the oral hygiene education in childhood as well as knowledge on brushing techniques according to the German Oral Health Survey (DMS; [27]). Furthermore, the relationship to the own mouth, sensations during oral hygiene and the perception and satisfaction with the own teeth were asked (self-developed questionnaires). In addition, established questionnaires on self-efficacy (SWE; [7,28]), dentist-related fears (DAS; [29]), personality profile (NEO-FFI; [30,31]), body perception (DKB-35; [32]) and the current psychological state (PHQ-D; [33]) were included.

The study has been registered in the German Clinical Trials Register (DRKS00012333). It was conducted in cooperation between the Department of Operative Dentistry and Periodontology and the Department of Psychosomatic Medicine and Psychotherapy of the Faculty of Medicine of the Medical Center—University of Freiburg. The clinical trial was conducted according to Good Clinical Practice (GCP E6 R2; [34]), in accordance with the ethical principles of the Helsinki Declaration [35] and approved by the ethics committee of the University Medical Center Freiburg on 3/28/2017 (application no. 59/17) (S1 Protocol). All volunteers received oral and written information about the procedures and the purpose of the study and gave written informed consent. All procedures were performed in the Department of Operative Dentistry and Periodontology of the Dental Clinic in Freiburg. Video-taping was performed through a two-way-mirror, collection of clinical data was performed at a dental unit and questionnaires were filled in at a desk in a closed room and in a quiet environment. The data collection was carried out in 2017. All data per participant were collected during a single appointment. All procedures and measures carried out were standardised and recorded in protocols (telephone protocol, meeting transcript and schedule, case report form). The study was carried out by two investigators (M.M. and S.S.), who were trained and calibrated.

The results are reported according to the STROBE Statement (S6 Table; [36]).

### Study collective, recruitment of volunteers and sample size calculation

The study included adult volunteers. All volunteers were students of the University of Freiburg without previous dental or psychological training or education. Inclusion criteria were written informed consent, age $\geq$ 18 years, routine use of a manual toothbrush for habitual oral hygiene, sufficient German language skills to answer the questionnaires written in German and closed dental arches. Exclusion criteria were fixed orthodontic appliances, removable dentures, routine use of an electric toothbrush and physical or mental impairment affecting oral hygiene measures.

Volunteers were recruited via advertisements on notice boards in the university and digitally via postings in university- and study-related social media groups. After replying by e-mail, interested parties were contacted by telephone by the investigators (M.M., S.S.). Using a standardised telephone protocol, the volunteers were informed about the purpose and procedure of the study, the inclusion and exclusion criteria were checked and, if possible, an

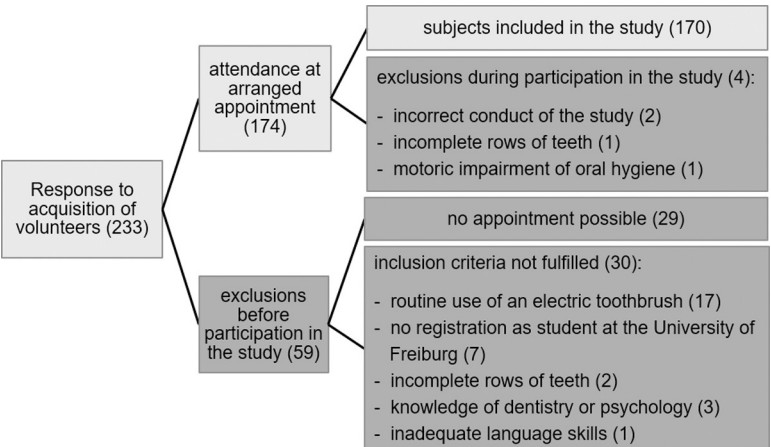

**Fig 1. Flow-chart of number and reasons for inclusion or exclusion of participants prior to and during the study.** Reasons and number (in brackets) of included or excluded participants.

appointment was made. After receiving oral and written information about the procedures and the purpose of the study and all volunteers gave written informed consent prior to participating in the study. For details of screening, inclusion and exclusion of participants see Fig 1.

Calculation of sample size for the whole study was performed by the Institute of Medical Biometry and Statistics, Freiburg, and was based on own data from a previous study [37] including a comparable population (students without knowledge in dentistry). From these data a toothbrushing systematics index was developed (TSI; [38]). This index was used in the present study as a measure for differences and changes in systematics of participants (data not shown). A maximum TSI score of 2 could be reached and described a perfectly executed systematic during cleaning-procedure of the teeth. The rationale behind the systematic was that all surfaces of the teeth should be equally reached during toothbrushing, all teeth and all surfaces should ideally be brushed for the same duration and the brush should be moved between the areas under investigation (in this case sextants) as few as possible. In the described previous study, participants, who were not previously instructed in using a brushing systematic, showed a mean index score of 1.2 with a MIN of 0.6, a MAX of 1.6 and a standard deviation (SD) of 0.3; this score therefore describes the degree of systematic which occurs during habitual (uninstructed) toothbrushing. A difference in TSI score between systematically and non-systematically habitual toothbrushing of 0.15 has been assumed as clinically relevant. For sample size calculation, a two-sided (non-directional) analysis was used using a t-test for independent samples. With $d = 0.15$, $SD = 0.3$, $\alpha = 0.05$ and $\beta = 0.1$ a sample size of 170 was determined.

## Procedures

**Questionnaire selection and data sources.** The study part presented here addressed the relationship between dental experiences, parental care and oral hygiene education and self-reported bushing behaviour. The participants answered standardized and newly developed questionnaires. The standardized questionnaires used were selected with regard to their reliability, validity, occurrence in the literature, simplicity and length of time required for completion. Standardized questionnaires were used to assess dental self-efficacy perception [7] and dental anxiety and dental fear (DAS—Dental Anxiety Scale [29]). The dental self-efficacy perception questionnaire provided results on the participants' confidence in brushing their teeth regularly, practising interdental hygiene or attending dental appointments in special situations

and despite barriers. Furthermore, various questionnaires on demographics, toothbrushing frequency, time and duration, use of oral hygiene products, frequency and intention of dental visits as well as taught toothbrushing techniques were taken from the Social Science Questionnaires of the German Oral Health Studies IV and V [39,40]. The detailed list of questions can be found in S1 Table. Additional dental-related questionnaires were newly developed by the principle investigators (A.Z. and N.S.). The self-developed questionnaires were tested in pilot runs and incomprehensible items were revised or removed. In these questionnaires, the relationship to one's own mouth, the attitude towards and sensations during oral hygiene, attitude towards one's teeth as well as perception and satisfaction with one's own teeth were assessed. Further questions dealt with dental-related experiences in childhood, oral hygiene education by parents, current and childhood experiences, sensations and emotions during dental visits and treatments. All answers were given on a five-level scale: the participant agrees "not", "barely", "in part", "largely" or "entirely" (S3 Table).

**Data handling, questionnaire evaluation and statistical analysis.** Apart from sections concerning life topics and situations participants had not experienced, the completeness of the answers to the questionnaires was checked by the investigators during the participation in the study and unintentionally missing information was added by the participant. For the parameters and factors presented and analysed here the data collection was complete, no missing data occurred.

The handwritten and marked answers to the questionnaires were transferred to a Microsoft Access database. Free text answers with similar content were grouped into subgroups and data correctness and plausibility were checked. The answers to the self-reported brushing behaviour and oral hygiene education are given only descriptively. For the other data, the statistical data analysis was performed with SAS (JMP; Version 13.2.1.; SAS Institute, Cary, NC, USA). The questionnaires DAS [41] and dental self-efficacy perception [7] were evaluated according to the descriptions in the underlying publications.

The newly composed dental questionnaires on "feelings towards oral hygiene", "motivation to brush teeth" and "feelings of pleasure and satisfaction with one's own teeth" (S4 Table, Questionnaires and newly composed questionnaires) were factorized into 5 independent factors by means of explorative factor analysis (Principal Component Analysis). The extracted factors were rotated using the Varimax method and adjusted for items with poor communalities (communalities < 0.2), multiple loads of factors (adjustment only for items with factor loads < 0.4) and eigenvalues ≥ 1. Eleven factors fulfilled these criteria but represented an overly differentiated picture with too few loading items. With five to seven factorial solutions the variance clarification (43–52%) and the internal consistency (Cronbach's α) improved with factors consisting of four or more items. A six-factor solution was chosen for the analysis of the study questionnaires. Further analyses and correlations with other results were only carried out with four of the six factors, which showed sufficient reliability (Cronbach's α > 0.5). Items that could not be clearly assigned to an independent factor or that were insufficiently loaded were not considered in the further correlations and evaluations (excluded Items: 4, 6, 9, 15, 16, 17, 18, 19, 23, and 29; (S3 and S4 Tables)).

The factor "attitude towards oral hygiene" (Cronbach's α = 0.8135) included 5 items ("I have a good feeling after brushing"; "brushing my teeth triggers a liberating feeling"; "I am looking forward to brushing my teeth"; "I find brushing my teeth boring" inverted; "brushing my teeth is an annoying duty" inverted). The factor "attitude towards one's teeth" (Cronbach's α = 0.7823) included 5 items ("My teeth feel good" inverted; "I can rely on my teeth" inverted; "I am satisfied with my teeth" inverted; "I am worried about being rejected because of my teeth"; "I am ashamed because of my teeth"). Factor "sense of care" (Cronbach's α = 0.6491) included 3 items on the topic of oral hygiene motivation ("I clean my teeth to make my mouth

cleaner"; "I brush my teeth to prevent bad breath"; "Well-groomed teeth are part of a well-groomed appearance"). The factor "self-inspection of one's teeth" (Cronbach's α = 0.5485) summarised 3 items treating actions for controlling one's own teeth and the use of professional help ("I notice changes in my teeth immediately"; "I regularly feel with my tongue whether everything is okay on my teeth"; "If I notice a dark spot on my teeth, I immediately make a dentist appointment").

The items relating to parental care for children's oral hygiene, current and childhood experiences with dentists and sensations during dental visits were also summarised into factors using rotating factor analyses and resulted in 3 factors with eigenvalues ≥ 1.

The factor "parental care" (Cronbach's α = 0.8245) summarised 5 items ("My parents took care of my oral hygiene"; "My parents motivated me with oral hygiene"; "My parents checked my teeth brushing"; "My parents regularly went to the dentist themselves"; "Brushing teeth was a natural part of my daily routine from an early age"). The factor "negative experiences in childhood" (Cronbach's α = 0.7283) included 3 items ("Early on (until puberty) I had a lot of dental problems—I always needed new fillings"; "When I was a child, I found visiting the dentist frightening"; "When I was little, I experienced dental treatment as painful"). And the factor "negative dentist experiences" (Cronbach's α = 0.8195) summarised 5 items ("I like to go to the dentist" inverted; "I find the visit to the dentist unpleasant, but meaningful"; "I already find the cleaning of my teeth unpleasant"; "My experiences so far have been bad"; "Only the thought of the dentist causes me a bad feeling").

The relations between the variables were examined with linear (bivariate) regressions, Pearson correlation coefficients or analysis of variance for differences. Only those factors are mentioned that contribute at least 5% to the explanation of the results ($r^2 \geq 0.05$). In order to derive the possible direction of the correlations, the r-value will be presented instead of $r^2$. The remaining data is only presented descriptively. The significance level α was set at 0.05 ($p \leq 0.05$). In the results presentation of the questionnaires with Likert scale, the two external evaluations ("largely" and "completely", "not" and "hardly") were summarised and are presented in percent.

## Results

Out of the 170 included participants, 119 were female and 51 male. The average age of the participants was 23.1 years (standard deviation 3.8; MIN: 18; MAX: 52). One hundred and fifty two participants grew up in Germany. Eighteen stated other countries of origin, 54.0% of those had lived in Germany for 2 years or longer. Only minor gender related differences were found; therefore, only in case of a significant impact of gender the differences are mentioned. Detailed results of questionnaires are given in S5 Table.

### Self-reported oral hygiene education

84.7% of the participants stated that they had been taught a toothbrushing technique at least once, 4.1% received no instruction and 11.2% were not sure (see Supporting Information: S2 Table Q1). More than half of those who were taught a toothbrushing technique named "individual prophylactic measures" (dentist) and more than one third "group prophylactic measures" (dentist in school/kindergarten) and parents/family members as mediating authorities (multiple answers possible) (S2 Table Q2).

### Self-reported brushing behaviour

Regarding brushing frequency, 81.8% brushed their teeth twice a day, 10.0% brushed at least 3 times per day, 7.6% once a day and 0.6% only several times a week (S2 Table Q4). The preferred time point of brushing can be seen in Fig 2 (S2 Table Q5).

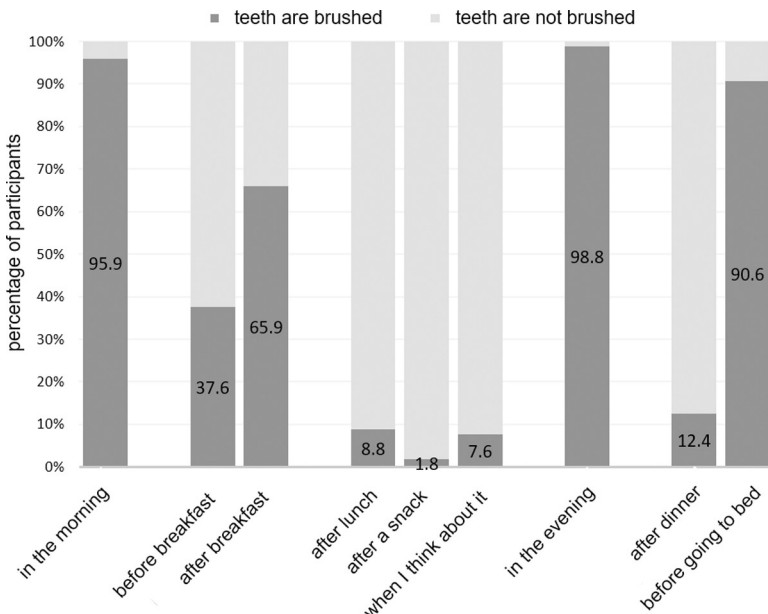

**Fig 2. Preferred time point of brushing.** Percentage of participants who brushed their teeth a certain points during the day (dark grey) or did not (light grey).

Self-estimated brushing duration was 1 min (9.4%), 1.5 min (11.8%), 2 min (48.2%), 3 min (27.6%) and longer than 3 min (2.9%) (S2 Table Q6).

## Self-reported dental visits

75.3% visited a dentist within the last 12 months, 17.1% within the last 2 years and 5.9% within the last 5 years; 1.2% could not answer the question and only 0.6% had never been to a dentist (S2 Table Q8). Regular control examinations at the dentist were perceived by 88.8% of the participants regularly (64.1%) or sometimes (24.7%); 8.2% only went to the dentist when they had pain; 2.4% could not answer the question and 0.6% had never been to a dentist (S2 Table Q9).

## Perception of one's own teeth and attitude towards oral hygiene

**Perception and value of one's own teeth.** More than two thirds of the participants had a positive attitude towards their own teeth ("I am satisfied with my teeth" "largely"—"entirely" 74.7% (see Supporting Information S5 Table: Domain (D)3 Item (I)21); "My teeth feel good" 84.2% (S5 Table D3 I22); "I can rely on my teeth" 77.0% (S5 Table D3 I24)). Accordingly, hardly any participant had had any experience with negative external effects of their teeth ("I'm ashamed of my teeth" "not"–"barely" 0.6% (S5 Table D3 I33); "I'm worried about being rejected because of my teeth" 1.2% (S5 Table D3 I32); "My partnership suffers under my teeth" 0% (S5 Table D3 I31)). Just as few participants had previously been asked by others about their own bad breath ("I have been approached about my bad breath several times" 0.6%) (S5 Table D3 I30), but almost half were afraid of it ("I'm worried about having bad breath" 42.9%) (S5 Table D3 I28).

Their own teeth were important for more than two thirds of the participants ("Teeth have always been important to me" 71.2%) (S5 Table D2 I8) and a tooth gap would be relevant for 86.5% ("I don't care whether I have a tooth gap or not" "not"- "barely") (S5 Table D3 I29).

**Oral health self-efficacy and oral hygiene motivation.** More than half of the test participants stated that they examined their own teeth ("I regularly check my teeth in the mirror"

55.3%; "I regularly feel with my tongue if my teeth are fine" 56.5%) (S5 Table D3 I23;I27) and would visit a dentist if they noticed anything irregular ("When I notice a dark spot on my teeth, I immediately make a dental appointment" 48.2%) (S5 Table D3 I26). Most participants were aware of the importance of brushing their teeth ("I don't really care about brushing my teeth" 5.9%) (S5 Table D1 I4).The respondents had high dental self-efficacy expectations and 91.1% believed that they could do a lot themselves to keep their teeth healthy ("I can do a lot myself to keep my teeth healthy" (S5 Table D2 I16)). The motivation for daily oral hygiene was focused on aspects of aesthetics ("I brush my teeth so that they look beautiful" 97.1% (S5 Table D2 I9), "Well-groomed teeth a part of a well-groomed appearance" 91.7% (S5 Table D2 I20)), general personal hygiene ("I clean my teeth so that the mouth gets cleaner" 81.8% (S5 Table D2 I13; "I brush my teeth so that I don't get any bad breath" 84.7% (S5 D2 I14)) and keeping their teeth healthy ("I brush my teeth to keep them healthy/to not get any tooth decay" 97.1% (S5 Table D2 I10)). Less relevant were the motivations by dentists ("My dentist has motivated me to brush my teeth" 25.9% (S5 Table D2 I15)), the desire to avoid complaints by the dentist ("I brush my teeth, so that my dentist doesn't find anything on my next visit" 35.3% (S5 Table D2 I11)) or an unspecific, socially ruled sense of duty ("I brush my teeth because one should do it just like that (out of a sense of duty)" 21.2% (S5 Table D2 I12)). Oral hygiene motivations, such as the desire for aesthetically pleasing teeth or a general or mouth-related desire for health, were grouped together under the factor "sense of care", which was the only factor showing significant differences between the sexes. Women showed significantly higher values for this factor (single factorial variance analysis: r = 0.267, p ≤ 0.001).

**Perception and Implementation of toothbrushing performance.** For most of the participants, the brushing procedure was associated with a good ("I have a good feeling after cleaning" 94.7% (S5 Table D1 I1)) or liberating ("Brushing my teeth releases a liberating feeling in me" 62.9% (S5 Table D1 I2)) feeling after brushing. At the same time, the process of toothbrushing was perceived as rather annoying ("Brushing my teeth is an annoying duty for me" 67.6% (S5 Table D1 I7)) or boring ("Toothbrushing is boring" 38.3% (S5 Table D1 I5)). Only 27.6% looked forward to brushing their teeth ("I look forward to brushing my teeth" (S5 Table D1 I3)) and 20.6% of the participants said they did other things while brushing ("I do something else while brushing my teeth" (S5 Table D1 I6)).

## Parental influence and experiences with dentists

**Influence of parental guidance.** Most of the participants were supervised ("My parents have taken care of my oral hygiene" 88.3% (S5 Table D5 I40)) and motivated ("My parents motivated my oral hygiene" 81.2% (S5 Table D5 I41)) by their parents regarding daily oral hygiene. In more than half of the participants, the parents had controlled the children's toothbrushing ("My parents checked my toothbrushing" 56.4% (S5 Table D5 I42)). For 86.4% of the participants, toothbrushing was a regular part of their daily routine from early childhood ("Brushing my teeth was a natural part of my daily routine from an early age on" (S5 Table D5 I44). In almost none of the participants, parental dental fears were transmitted to the children ("My parents transferred their fear of dentists onto me" 0.0% (S5 Table D5 I46)) and almost three quarters of the participants' parents went to the dentist regularly themselves ("My parents regularly went to the dentist themselves" 74.1% (S5 Table D5 I43)).

**Influence of previous dental experiences and attitude towards dental visits.** While almost half of the participants visited an orthodontist frequently in their childhood, appointments for dental restorations were less frequent ("When I was a child or teenager, I often had to go see an orthodontist" 48.9% (S5 Table D5 I47); constant need for new fillings 10.0% (S5 Table D5 I48)). Rarely had the participants had painful experiences with dentists in their

childhood or found the dental visits frightening ("As a child, I felt frightened visiting the dentist" 14.1% (S5 D5 I49); "When I was little, I experienced dental treatment as painful" 10.0% (S5 Table D5 I50)).

Up to the date of data collection, 26.5% of the participants had only been to the dentist for dental check-ups and not for invasive dental treatments ("I have always been at dental check-ups, nothing's ever had to be done" (S5 Table D5 I43)). Check-ups at the dentist were attended by 88.8% of the participants, on a regular basis (64.1%) or sometimes (24.7%). Only 8.2% of the respondents stated that they only went to the dentist if they experienced pain. The majority of the participants associated the visit to the dentist with neutral up to positive feelings ("I like going to the dentist" 43.6% (S5 Table D5 I34); "I find the visit to the dentist unpleasant but useful" 40.5% (S5 Table D5 I36); "My experiences so far have been bad" 3.0% (S5 D5 I38); "Even the professional cleaning is unpleasant" 10.6% (S5 Table D5 I37); "Just the thought of going to the dentist gives me a bad feeling" 4.7% (S5 Table D5 I39)).

### Dental anxiety and dental self-efficacy perception

**Dental anxiety.** Based on the answers to the DAS, the participants were divided into groups with regard to dental anxiety. Most of the participants fell into the "low anxiety"group (88.0%) and 12.0% were considered "medium anxious". Thus, only one participant classified as "strongly anxious" in dental situations. While many dental-related situations caused no or only minor anxiety symptoms, the mere thought of using a drill caused tension in 31.0% and strong anxiety in 11.0%.

**Dental self-efficacy perception.** The participants reported high self-efficacy perceptions with regard to toothbrushing. Situations or obstacles such as free time, illness and stress did not reduce the confidence in brushing the teeth in almost all participants (92.4–98.3%). The greatest inhibition for toothbrushing was strong tiredness. Only 80.0% were still completely or largely confident in brushing their teeth despite this fatigue. Products for interdental hygiene were used by only 65.9% of the participants. Of these, 64.7–70.6% were fairly or completely confident in using them, even if no visit to the dentist was scheduled, they were tired or very busy, had headaches or feelings of illness. Concerning dental visits, 64.7–70.6% were confident to go to the dentist in the recommended interval, even if the dentist did not remind them, if they had financial shortages or were not experiencing any discomfort. Not being able to make an appointment with a known dentist or being very busy reduced the confidence to make regular dental appointments by 47.1–58.3%. The most common reason for the decrease in confidence in making dental appointments was when the participants had had unpleasant experiences or were afraid of painful treatments (69.4–70.6%).

### Relationships between various questionnaires and factors

We explored the relationship of recent attitudes and emotions towards teeth and protective oral behaviour with former events (history and biography), dental anxiety and aversive experiences in dental practices. We found moderate negative correlations with recent negative experiences in dental practices, negative childhood experiences, and Dental Anxiety (DAS) (Fig 3)). Interestingly, the DAS correlated highly with recent negative experiences with dentists as well as during childhood (r = 0.694; 0.466, p ≤ 0.001; 0.001). Current and childhood negative experiences also correlated significantly with each other (r = 0.521, p ≤ 0.001). The DAS was significantly negatively correlated with attitudes towards oral hygiene and one's own teeth and self-control of teeth (r = -0.294; -0.271; -0.226, p ≤ 0.001; 0.001; 0.01). Recent negative experiences and sensations in dental practices showed significant negative correlations with attitudes towards oral hygiene and own teeth and teeth self-control (r = -0.490; -0.304; -0.236,

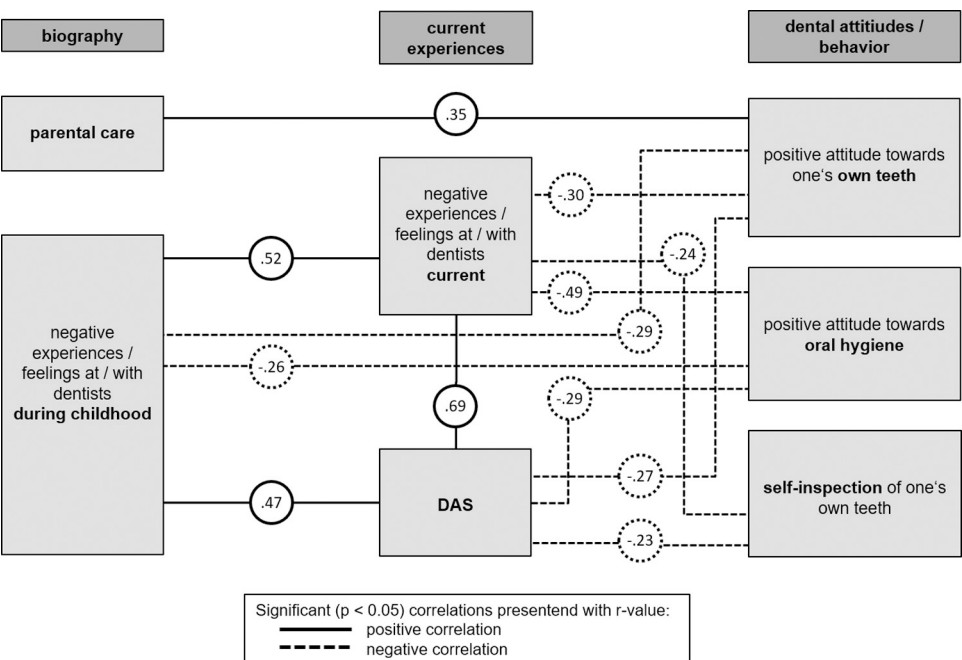

**Fig 3. Significant correlations (r-value) between environmental factors and the independent factors of oral hygiene behaviour.** Significant (p ≤ 0.01) positive (full line) and negative (dotted line) correlations between environmental factors (parental care; negative experiences / feelings at / with dentists currently and during childhood; dental anxiety) and the 4 independent oral hygiene behaviour factors (factor 1 "attitude towards oral hygiene" (including sensations towards and during oral hygiene); factor 2 "attitude towards one's teeth" (including perception of one's own teeth); factor 4 "self-inspection of one's teeth" (including actions for controlling one's own teeth and the use of professional help). Only those results are presented exceeding the threshold of effect size ($r^2 > 5\%$).

p ≤ 0.001; 0.001; 0.01). Negative childhood dental experiences also correlated with attitudes towards oral hygiene and one's own teeth (r = -0.261; -0.287, p ≤ 0.001; 0.001). On the other hand, parental care was correlated with positive attitudes towards own teeth (r = 0.349, p ≤ 0.001; Fig 3)).

## Discussion

Oral health behaviour is essential for maintaining the integrity of the teeth and the oral cavity as a whole. The importance and actual implementation of oral hygiene and other oral health maintenance measures should be conveyed by dental professionals as well as in the private environment. Unfortunately, there are great differences in oral health and oral hygiene depending on the social environment.

Regarding oral hygiene, the study sample showed slightly better results than comparable surveys of the general population. In the DMS, 80–83% of the 12-year-olds and 35-44-year-olds said that they usually brush their teeth at least twice a day [39] and 31–45% showed a toothbrushing pattern defined as "rather good". This meant that teeth were brushed at least twice a day for 2 minutes after meals or before going to bed. In this study, 91% of the respondents, with an average age of 24 years, reported brushing twice a day or more and 78% reported a self-estimated brushing duration of more than 2 minutes. The most frequent times for oral hygiene measures were reported to be in the morning after breakfast and in the evening before going to bed. The higher frequency of toothbrushing could be due to the participant selection of university students only. In the DMS, people from all social classes are interviewed. The rather high social status in this collective could have had a positive influence on the

frequency of oral hygiene measures [42,43]. Consequently, one could assume that a high social status also affects dental visit behaviour. This could explain the lower incidence of complaint-oriented dental visit behaviour in the current study compared to studies with surveys of representative groups. In the DMS V, 18% of 12-year-olds and 22–39% of 35-44-year-olds stated that they only went to the dentist if they had dental problems or pains [39]. Positive behaviours, such as keeping regular dental appointments, were found in most of the participants and regular check-ups at the dentist were perceived with similar frequency in both studies (present study: 64%; DMS V: 12-year-olds and 35-44-year-olds: 61–82% [44]).

Toothbrushing was, like other body hygiene tasks, viewed as a natural part of daily routine [45,46]. In most participants, the motivation for toothbrushing was based on intrinsic or internalised socially conditioned motives of aesthetics [47], general body hygiene and health maintenance. The factor "sense of care" summarised toothbrushing motivations. It was the only factor investigated that showed significant differences between the sexes and women achieved significantly higher scores than men. Differences in health and prevention-focused behaviour between the sexes are also known in other areas. For example, women generally make more frequent use of health services, eat better, are less likely to be overweight, smoke less and drink less alcohol. [48,49]. The adoption of oral hygiene measures and the internalisation of the motivation for them can be seen as a successful inclusion of dental prevention measures in social norms [50,51].

Even if the participants perceived the process of toothbrushing itself as boring or annoying, most of the participants cared about their teeth and were aware of the relevance of oral hygiene measures. Accordingly, the feeling after brushing was good or even liberating and the participants demonstrated high self-efficacy perceptions with regard to brushing despite difficulties and unfamiliar situations.

Positive effects of strong self-efficacy perceptions on oral health behaviour, like higher frequencies of toothbrushing and dental visits, are known from literature [5,52,53]. In this study, dental self-efficacy perceptions also correlated with a positive attitude towards toothbrushing and with participants' independent control of their teeth. The dental self-efficacy perception was reduced by negative dental experiences and dental anxiety but also positively influenced by increased parental care. Dental self-efficacy perception seems to be sensitive to external influences and should be considered for modulations of oral health behaviour when developing new oral hygiene interventions [3,15]. These results clearly show the importance of parental influence and the need for parents to be intensively involved in the oral hygiene instruction of their children at a very early age.

Only few participants had had recent or childhood negative experiences during dental visits. However, if they occurred, they correlated negatively with the perception of one's own teeth and the attitude towards oral hygiene. Negative experiences reduced not only the control of one's own teeth at home, but also the participants' perception of self-efficacy in regularly visiting the dentist. As expected, current and childhood negative experiences at dentist visits correlated with the existence and development of dental anxiety. Participants with higher DAS scores took their teeth as being more unreliable and less pleasant, were more ashamed of their teeth and afraid of being rejected because of them [54]. In terms of oral health measures, dental fears were associated with a negative attitude towards brushing and reduced control of their own teeth. The links between dental fears and poor oral health and hygiene can be confirmed [19,20,55]. These results turned out that child-oriented and patient-centred dentistry and sufficient analgesia during patients' treatment could contribute to a better relationship to the own oral cavity and possibly also to a higher motivation for oral hygiene procedures. Negative and painful events should therefore be prevented as far as possible when treating children in order

to counteract the development of dental fears. In the case of existing fears, in addition to dental anxiety therapy the dentist should also focus on establishing sufficient oral health behaviour.

Existing parental dental anxieties were not transferred to the children in the study, but a greater care of the parents for their children's oral health correlated with children's positive perception of their own teeth. As children often learn behaviours by imitating their parents, these can be passed on through generations [21,56,57]. Studies have shown that the oral health-related behaviour of parents and guardians, such as frequency of toothbrushing, fear of dentists, tooth visiting behaviour, self-perception of oral hygiene, experience of tooth decay, attitude towards oral hygiene, oral health-related self-efficacy perception and maternal care, influenced the oral hygiene behaviour of children with regard to toothbrushing frequency, dental appointments and fear of dentists [21–24]. Attitudes and beliefs about oral hygiene and dental visits also appear to be influenced by the parents' previous experiences [58–60]. Therefore, parents in particular should be included in oral hygiene strategies, since the motivation for oral hygiene and its daily implementation was initiated and controlled by the parents in the majority of the participants [61].

It is often stated that oral health behaviour could be affected by a variety of factors such as age, gender, social norms, social status or level of education [44]. Therefore, very precise inclusion and exclusion criteria were defined, in order to obtain a homogenous group of participants, in which the influences of psychological variables and person's individual experiences on oral health behaviour can be sufficiently investigated. Possible individual differences influencing behaviour should not be obscured by environmental influences, motor limitations or physical obstacles in the oral cavity and prior dental or psychological knowledge about desired answers or outcomes. Therefore, these criteria were also considered as exclusion criteria. Variations in age and educational level were minimized by the exclusive recruitment of university students. The influence of previous dental and psychological knowledge on the answers to the questionnaires was controlled by excluding students and professionals from these fields of work. Considering the lack of significant differences between the sexes, the increased participation of female participants did not prove to be problematic, at least in the cohort investigated in the study. Even if the study collective was not a representative sample of the general population, the results could nevertheless provide indications of certain correlations and thus may point the way for further, more comprehensive investigations.

The study was carried out by two investigators (M.M. and S.S.), who were trained and calibrated to reduce possible investigator-centered bias. Still, the study outcome could have been biased by the proband's participation in the study itself (Hawthorne effect). Another source of bias is the fact that the participants volunteered for the study, which may have led to a selection of the more motivated participants.

The standardized questionnaires used to record dental self-efficacy perception (dSEP [7]) and dental anxiety (DAS [41]; German adaptation [29]) are validated and reliable. They have been clinically tested and are considered as standard for the collection of the respective parameters in the literature. In accordance with the results of the Dental Anxiety Scale (DAS), only one participant fell into the group defined as "strongly anxious". Thus, only conclusions about a non-anxious collective can be drawn from this study. The questionnaires on childhood experience, dental experience and oral hygiene education were newly developed for the study and are therefore not validated. Since no templates existed for these questionnaires, rotating factor analyses were used to identify independent factors that could be correlated with other data collected in the study. The factors should explain the overall variance as widely as possible (about 50%) and not under- or over-differentiate the items. For this purpose, the factors had to have eigenvalues greater than 1 and be constant in themselves (Cronbach's $\alpha > 0.5$) in order to categorize and summarise the data collected from the items. The Cronbach for "self-inspection of

one's teeth" is quite low, since the scale consisted only of three items, which is certainly an obstacle to internal consistency. The results obtained with this scale must be interpreted with caution, and it should be revised and enlarged in future investigations.

In conclusion, the present study clearly shows, at least for the healthy group studied, that both experience and education in dental health during childhood significantly influence the lifelong relationship with the oral cavity. Therefore, parents have to be included into oral hygiene education as early as possible. They serve as role models and children learn many behaviours by simply imitating them. Parents should be educated thoroughly by dentists about oral hygiene measures and should be more closely involved in the process of oral hygiene instruction. Furthermore, negative experiences at a dentist and the resulting anxiety and fear resulted not only in avoidance behaviour regarding visits to the dentist, but also in a poorer attitude towards oral hygiene measures and a decreased toothbrushing motivation. As consequence, early negative experiences at dentists should be avoided best possible. On the other hand, the participants showed a high self-expectation but also high motivation for oral hygiene, which can clearly be judged as a success of the current preventive programmes in Germany. Further research should address a wider population in order to increase generalisability of the results.

## Supporting information

**S1 Table. Questions on toothbrushing frequency, time, duration, use of oral hygiene products, frequency and intention of dental visits, taught toothbrushing techniques taken from the German Oral Health Studies IV and V (translated into English).** Multiple answers possible.
(DOCX)

**S2 Table. Frequency of responses in percentage given to questionaire on toothbrushing frequency, time, duration, use of oral hygiene products, frequency and intention of dental visits, taught toothbrushing techniques taken from the German Oral Health Studies IV and V (translated into English).**
(DOCX)

**S3 Table. Newly composed questionnaires (translated into English).**
(DOCX)

**S4 Table. Rotated factor loads and communalities of the items of the newly developed questionnaires.** Items with grey background displayed too poor communalities or were not found in the chosen six-factorial solution and were not included in the further analyses; communalities in bold letters indicate that they belong to the respective factor.
(DOCX)

**S5 Table. Percentage of responses to the newly developed questionnaires.** Percentage data of the frequency of answering the items with the answer options "not", "barely", "in part", "largely" or "entirely".
(DOCX)

**S6 Table. Information in accordance to STROBE statement.**
(DOCX)

**S1 Protocol. Study protocol of underlying study.**
(PDF)

## Acknowledgments

We thank all volunteers for their participation in the study.

## Author Contributions

**Conceptualization:** Almut Zeeck, Nadine Schlueter.

**Data curation:** Maxi Mueller, Sarah Schorle, Armin Hartmann, Almut Zeeck, Nadine Schlueter.

**Formal analysis:** Armin Hartmann.

**Investigation:** Maxi Mueller, Sarah Schorle.

**Methodology:** Armin Hartmann, Almut Zeeck, Nadine Schlueter.

**Validation:** Armin Hartmann, Almut Zeeck, Nadine Schlueter.

**Visualization:** Maxi Mueller.

**Writing – original draft:** Maxi Mueller, Sarah Schorle, Armin Hartmann, Almut Zeeck, Nadine Schlueter.

**Writing – review & editing:** Maxi Mueller, Sarah Schorle, Kirstin Vach, Armin Hartmann, Almut Zeeck, Nadine Schlueter.

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
