## [Decision Letter · Decision Letter 0]

19 May 2021

PONE-D-20-29851

Relation between dental experiences, oral hygiene education and individual psychological variables and self-reported oral hygiene behaviour

PLOS ONE

Dear Dr. Mueller,

Thank you for submitting your manuscript to PLOS ONE. After careful consideration, we feel that it has merit but does not fully meet PLOS ONE’s publication criteria as it currently stands. Therefore, we invite you to submit a revised version of the manuscript that addresses the points raised during the review process.

Please ensure that you address the comments raised by the reviewers regarding the explanation of your study rationale and design, and respond to the presentational points indicated in their reports.

We look forward to receiving your revised manuscript.

Kind regards,

Jamie Males

Staff Editor

PLOS ONE

Journal Requirements:

2. Please include captions for ALL your Supporting Information files at the end of your manuscript, and update any in-text citations to match accordingly. Please see our Supporting Information guidelines for more information: http://journals.plos.org/plosone/s/supporting-information.

Reviewers' comments:

Reviewer's Responses to Questions

**Comments to the Author**

1. Is the manuscript technically sound, and do the data support the conclusions?

Reviewer #1: Partly

Reviewer #2: Partly

2. Has the statistical analysis been performed appropriately and rigorously? 

Reviewer #1: No

Reviewer #2: I Don't Know

3. Have the authors made all data underlying the findings in their manuscript fully available?

Reviewer #1: Yes

Reviewer #2: No

4. Is the manuscript presented in an intelligible fashion and written in standard English?

Reviewer #1: Yes

Reviewer #2: Yes

5. Review Comments to the Author

Reviewer #1: The study aims to assess the relationship between dental experiences, oral hygiene education and individual psychological variables and self-reported oral hygiene behaviour.

The manuscript can be further improved based on the following comments.

The following sentence requires improvement/revision.

i) The title of the manuscript.

ii) Abstract - the sentence ‘The results of the questionnaires were correlated.

iii) Line 146, the sentence ‘students without relation to dentistry’.

iv) Line 236, 237.

v) The word relation to be replaced with relationship where applicable.

Line 127, the language used including the questionnaires language version to be stated. i.e. German language.

Line 152, 1 or 2-tailed test to be stated.

Line 166 and S1 Table, questionnaires or questions?

Line 169, 170, the tables to be cited.

Line 177, the Likert scale which was mentioned is different from the questionnaire (S2 Table) e.g. partly vs in part.

Statistical software and level of accepted significance to be stated.

Results

Line 243, sd to be labelled/highlighted.

Line 284, ensure the exact words from the table are used when describing in the text in the results section e.g. aesthetics (teeth should look nice 71%).

For percentage figures which were presented in the text results section, there were discrepancies in terms of the combination to derive the percentage figures whether using the sum of [in part, largely and entirely] or using sum of [largely and entirely]. This needs to be standardized.

Line 338-348, since there are quite a number of variables involved and possibility of the presence of mediator(s), moderator(s) effect and to determine the pathway, the analysis could be explored using Structural Equation Modelling (SEM) approach following EFA.

At least 1 decimal point to be used when displaying percentage figures (in the text results section and Figure 2).

S1 Table, typo Englisch.

S2 Table, the statement is true to be omitted. The translated English version for the Likert scale words are not commonly used in Likert scale e.g. could have used ''Strongly Agree, Agree, Mixed/Neutral, Disagree, Strongly Disagree'. Questionnaire (without s) to be used since all questions are considered under one questionnaire and also all questions numbered in ascending order according to domains.

S4 Table, the figures with comma to be replaced with dot.

For the benefit of reader(s), all the percentage figures which were written in the text in the results section to be cited with table number and domain names/section.

Not all references are formatted according to PLoS One format.

Reviewer #2: Oral health is a worldwide public health priority and understanding the factors that relate to oral health behaviours is vital to explore in order to best direct future intervention strategies. This research is therefore important, however, there a number of ways the manuscript could be strengthened. Please find detailed comments below.

General

• The title is quite wordy, and I wonder if there is scope to make this pithier and more concise

• ‘Participants’ rather than ‘subjects’ or ‘test persons’

• Check for grammar

Abstract

• The opening sentence is rather vague and doesn’t quite flow. It would benefit from some re-phrasing.

Introduction

• Oral health behaviour encompasses a wide range of varied behaviours, which needs to be reflected, as at present the description is quite vague and doesn’t reflect the complexity of behaviour.

• The research covers a broad range of variables that could easily be considered within their own right, and it’s not clear why you have chosen to look at these variables and why you have chosen to look at them all together. The introduction needs to tell a story that justifies the current research.

• Aim and objectives are missing.

Methods

• Issues with bias and sample would be better placed in a strengths and limitations section within the discussion section.

• What is the justification for the inclusion/exclusion criteria?

• “…from this time of inclusion they were called participants or subjects” – why?

• “From these data a 147 toothbrushing systematics index was developed (TSI; [38]). This index was used in the present study as a measure for differences and changes in systematics of participants (data not shown). A maximum TSI score of 2 could be reached. Non-instructed participants had a mean 150 index score of 1.2 with a MIN of 0.6, a MAX of 1.6 and a SD of 0.3.” - This needs further explanation to understand what it means. What are systematics? Also, what is the relevance of ‘non-instructed’?

• ‘Demographics’ rather than ‘demography’

• “The completeness of the answers to the questionnaires was checked by the investigators during the participation in the study and missing information was added by the participant” – does this mean participants were not able to leave questions unanswered if they so wished?

• The Cronbach for "self-inspection of one’s teeth" is quite low

Results

• For clarity I would recommend breaking down the sub-headings further

Discussion

• The opening sentence is rather vague and doesn’t quite flow. It would benefit from some re-phrasing.

• This section would benefit from some restructuring by starting first with the findings from your research and how it relates to the wider literature and then moving onto critically appraising the study.

• Referencing style changes in this section-use consistent format throughout

• Conclusion needs strengthening

6. PLOS authors have the option to publish the peer review history of their article (what does this mean?). If published, this will include your full peer review and any attached files.

Reviewer #1: No

Reviewer #2: No

---

## [Author Response · Author response to Decision Letter 0]

5 Jul 2021

Dear Dr. Heber, dear Reviewers,

We thank you very much for the thorough revision of our manuscript and the constructive feedback. We revised the whole manuscript carefully with respect to the reviewers’ comments. All changes are marked in yellow within the manuscript. Please see below for a point-by-point response to the reviewers comments.

Reviewer #1: 

The study aims to assess the relationship between dental experiences, oral hygiene education and individual psychological variables and self-reported oral hygiene behaviour.

The manuscript can be further improved based on the following comments.

The following sentence requires improvement/revision.

i) The title of the manuscript.

We shortened the title in order to improve readability. The title is now:

“Relationship between dental experiences, oral hygiene education and self-reported oral hygiene behaviour”

ii) Abstract - the sentence ‘The results of the questionnaires were correlated.

The sentence has been altered to: „The results of the questionnaires were correlated by means of linear regressions.“

iii) Line 146, the sentence ‘students without relation to dentistry’.

The sentence was altered to: „students without knowledge in dentistry“ (Line 147)

iv) Line 236, 237.

The sentence “The factors thus constructed as well as the results of the standardized psychological questionnaires were correlated via linear (bivariate) regressions and variance analyses with each other.” has been revised to „The constructed factors on attitudes toward oral hygiene and own teeth, oral hygiene motivation, self-inspection of own teeth, and parental oral health care were examined with linear (bivariate) regressions and analyses of variance for correlations among them and for correlations with the results of the standardized psychological questionnaires on dental anxiety and dental self-efficacy perceptions."

v) The word relation to be replaced with relationship where applicable.

The word “relation” has been replaced by “relationship“ in the title as well as Line 84, 158, 468

- Line 127, the language used including the questionnaires language version to be stated. i.e. German language. 

The language has been added and the sentence now reads: „…sufficient German language skills to answer the questionnaires written in German…“

- Line 152, 1 or 2-tailed test to be stated.

For sample size calculation a two-sided (non-directional) analysis was used using a t-test for independent samples. Information on this has been added to the text.

- Line 166 and S1 Table, questionnaires or questions?

We meant “questions”. We changed it in the text and in S1

- Line 169, 170, the tables to be cited.

The cited tables concerning the followed DMS questionnaires have been added to the underlying literature: 

• DMS V Chapter 8: Appendix 8.6.1-8.6.2.: Social science questionnaire for children; Social Science Questionnaire for Younger Adults pp. 150-184; citation altered accordingly

• DMS IV Chapter 8: Appendix 8.5.1-8.5.2.: Social science questionnaire for children; Social Science Questionnaire for Adults pp. 119-139; citation altered accordingly

- Line 177, the Likert scale which was mentioned is different from the questionnaire (S2 Table) e.g. partly vs in part.

Many thanks for this annotation of this incongruence. We used now in each case: “not, barely, in part, largely and entirely”.

- Statistical software and level of accepted significance to be stated.

The SAS Version has been added (13.2.1.); the level of accepted significance (r² ≥ 0.05) is now stated in Line 242.

- Results - Line 243, sd to be labelled/highlighted.

The sentence has been modified: “The average age of the subjects was 23.1 years (standard deviation 3.8; min: 18; max: 52)”.

- Line 284, ensure the exact words from the table are used when describing in the text in the results section e.g. aesthetics (teeth should look nice 71%).

Exact wording was adopted or inserted from questionnaires.

- For percentage figures which were presented in the text results section, there were discrepancies in terms of the combination to derive the percentage figures whether using the sum of [in part, largely and entirely] or using sum of [largely and entirely]. This needs to be standardized.

Many thanks for this important point. We have standardized the percentage summarizing and now, only the combination of answers „largely“ and „entirely“ or „not“ and „barely“ are given.

- Line 338-348, since there are quite a number of variables involved and possibility of the presence of mediator(s), moderator(s) effect and to determine the pathway, the analysis could be explored using Structural Equation Modelling (SEM) approach following EFA.

It is a valuable suggestion to summarize the correlations in a SEM. A SEM and it's visualizations would make the realtionships and implicit temporal dependencies easier to understand. Yet, the reported correlations refer to three "dependent" variables (attitudes towards oral hygiene and one's own teeth and self-control of teeth) which would mean either constructing three SEMs or creating one latent variable "positive attitudes and behavior" from the three dependent variables. In our view, both alternatives have severe downsides: (1) Reporting three models seems too space consuming and theoretically unjustified; (2) Creating one latent variable has no apriori foundation in hypotheses or theory. Therefore we did not add a SEM to the manuscript, but we did rewrite the repsective paragraph for improved structure and clarity and added a small table to improve readability of the correlations.

- At least 1 decimal point to be used when displaying percentage figures (in the text results section and Figure 2).

We added one decimal throughout the results section and in Fig 2.

- S1 Table, typo Englisch.

This typo and other typos have been corrected.

- S2 Table, the statement is true to be omitted. The translated English version for the Likert scale words are not commonly used in Likert scale e.g. could have used ''Strongly Agree, Agree, Mixed/Neutral, Disagree, Strongly Disagree'. Questionnaire (without s) to be used since all questions are considered under one questionnaire and also all questions numbered in ascending order according to domains.

The word has been changed to „Questionnaire“. 

Regarding the scale: Since newly developed questionnaires were used, we translated the German wording as correct as possible. The most accurate translation from the phrasing we used in the German questionnaire were the named ones. An adaption at this time to the common Likert scale would bear the risk of changing the sense and following the participants’ intention of answers. However, this is a valuable point for further research that we will keep in mind.

- S4 Table, the figures with comma to be replaced with dot.

According to the recommendation, the comma has been replaced with dot in S3 and S4.

- For the benefit of reader(s), all the percentage figures which were written in the text in the results section to be cited with table number and domain names/section.

For more clarity, we have structured the results section and added subheadings to the section. 

- Not all references are formatted according to PLoS One format.

All references are now formatted according to the PLoS One format

Reviewer #2: 

Oral health is a worldwide public health priority and understanding the factors that relate to oral health behaviours is vital to explore in order to best direct future intervention strategies. This research is therefore important, however, there a number of ways the manuscript could be strengthened. Please find detailed comments below.

General

- ‘Participants’ rather than ‘subjects’ or ‘test persons’

The terms “subject” and “test person” have been replaced by „participant“

- Check for grammar

The whole text has been read again and checked for typos, style and grammar. Respective parts have been corrected.

Abstract

- The opening sentence is rather vague and doesn’t quite flow. It would benefit from some re-phrasing.

We rephrased the first sentences: “Many preventive approaches in dentistry aim to improve oral health through behavioural instruction or intervention concerning oral health behaviour. However, it is still unknown which factors have the highest impact on oral health behaviours, such as toothbrushing or regular dental check-ups. Various external and internal individual factors such as education, experience with dentists or influence by parents could be relevant. Therefore, the present observational study investigated the influence of these factors on self-reported oral heath behaviour.”

Introduction

- Oral health behaviour encompasses a wide range of varied behaviours, which needs to be reflected, as at present the description is quite vague and doesn’t reflect the complexity of behaviour.

- The research covers a broad range of variables that could easily be considered within their own right, and it’s not clear why you have chosen to look at these variables and why you have chosen to look at them all together. The introduction needs to tell a story that justifies the current research.

We added some more information to the introduction beginning and end: “Over the last decades, prevention of oral diseases came more and more in focus as there is evidence that oral health is not only a matter of oral wellbeing and quality of life; it can also affect the health of the whole body. Therefore, it is important not only to maintain the chewing function of the teeth, but also to decrease inflammatory processes or pain and maintain the aesthetics for social and psychological reasons. Oral health behaviours summarise a wide range of measures designed to help preventing any alteration of oral health. In addition to the passive avoidance of harmful behaviour, such as smoking or the consumption of tooth-damaging foods, this also includes behavioural patterns that must be actively practised. Especially for active measures, their relevance must be conveyed to the patient and their implementation must be internalised so that they are maintained over the course of the patient's life. The most widespread and individually enforceable measures are oral hygiene measures such as regular and careful tooth brushing with a fluoride toothpaste and interdental cleaning, as well as dental check-ups and the according implementation of any necessary therapies. …“

“… This shows that a large and diverse spectrum of possible factors might be relevant influencing oral health behaviour. As mentioned, single factors have already been investigated regarding their role, however, the interaction of them have never been in the focus of research..”

- Aim and objectives are missing.

Aims and objectives have been added: “The aim of the study was to explore if and to what extent individual personality variables and experiences affect oral health behaviours. Here, a part of this study is presented. The study objectives of this study part were correlations between parameters of dental experiences, parental care and oral hygiene education with self-reported oral hygiene behaviours, such as toothbrushing frequency, the use of interdental hygiene aids, frequency and intention of dental visits and attitudes towards those.”

Methods

- Issues with bias and sample would be better placed in a strengths and limitations section within the discussion section.

The study was reported according to the STROBE statement. According to the checklist, the biases should be placed to the M&M section. However, we agree with the reviewer and moved the paragraph “The study was carried out by two investigators (M.M. and S.S.), who were trained and calibrated to reduce possible investigator-centered bias. Still, the study outcome could have been biased by the proband’s participation in the study itself (Hawthorne effect). Another source of bias is the fact that the participants volunteered for the study, which may have led to a selection of the more motivated participants.”, which is dealing with the biases, to the discussion section. 

- What is the justification for the inclusion/exclusion criteria?

Further explanation has been added to the discussion section: “Therefore, very precise inclusion and exclusion criteria were defined, in order to obtain a homogenous group of participants, in which the influences of psychological variables and person's individual experiences on oral health behaviour can be sufficiently investigated Possible individual differences influencing behaviour should not be obscured by environmental influences, motor limitations or physical obstacles in the oral cavity and prior dental or psychological knowledge about desired answers or outcomes. Therefore, these criteria were also considered as exclusion criteria..”

- “…from this time of inclusion they were called participants or subjects” – why?

We agree with the reviewer and have deleted the sentence.

- “From these data a 147 toothbrushing systematics index was developed (TSI; [38]). This index was used in the present study as a measure for differences and changes in systematics of participants (data not shown). A maximum TSI score of 2 could be reached. Non-instructed participants had a mean 150 index score of 1.2 with a MIN of 0.6, a MAX of 1.6 and a SD of 0.3.” - This needs further explanation to understand what it means. What are systematics? Also, what is the relevance of ‘non-instructed’?

We added some more information for clarity. The respective paragraph is now: “A maximum TSI score of 2 could be reached and described a perfectly executed systematic during cleaning-procedure of the teeth. The rationale behind the systematic was that all surfaces of the teeth should be equally reached during toothbrushing, all teeth and all surfaces should ideally be brushed for the same duration and the brush should be moved between the areas under investigation (in this case sextants) as few as possible. In the described previous study, participants, who were not previously instructed in using a brushing systematic, showed a mean index score of 1.2 with a MIN of 0.6, a MAX of 1.6 and a standard deviation (SD) of 0.3; this score therefore describes the degree of systematic, which occurs during habitual (uninstructed) toothbrushing. A difference in TSI score between systematically and non-systematically habitual toothbrushing of 0.15 has been assumed as clinically relevant. For sample size calculation, a two-sided (non-directional) analysis was used using a t-test for independent samples. With d = 0.15, SD = 0.3, α = 0.05 and β = 0.1 a sample size of 170 was determined. “ 

- ‘Demographics’ rather than ‘demography’

The word has changed as recommended.

- “The completeness of the answers to the questionnaires was checked by the investigators during the participation in the study and missing information was added by the participant” – does this mean participants were not able to leave questions unanswered if they so wished?

- Many thanks for mentioning this point. This was for sure not the case. The sentence has been corrected to: „Apart from sections concerning life topics and situations participants had not experienced, the completeness of the answers to the questionnaires was checked by the investigators during the participation in the study and unintentionally missing information was added by the participant.“

- The Cronbach for "self-inspection of one’s teeth" is quite low

We agree with the reviewer’s evaluation. This scale is very short with only three items, which is certainly an obstactle to internal consistency. The results obtained with this scale must be interpreted with caution, and it should be revised and enlarged in future investigations. We added this note to the discussion.

Results

- For clarity I would recommend breaking down the sub-headings further

Many thanks for this suggestion. We have added some more subheadings.

Discussion

- The opening sentence is rather vague and doesn’t quite flow. It would benefit from some re-phrasing.

We rephrased the sentence in order to improve readability: “Oral health behaviour is essential for maintaining the integrity of the teeth and the oral cavity as a whole. The importance and actual implementation of oral hygiene and other oral health maintenance measures should be conveyed by dental professionals as well as in the private environment. Unfortunately, there are great differences in oral health and oral hygiene depending on the social environment.”

- This section would benefit from some restructuring by starting first with the findings from your research and how it relates to the wider literature and then moving onto critically appraising the study.

Many thanks for this suggestion: We restructured the discussion section as follows: a) Findings; b) Relation to wider literature; c) critical appraise

- Referencing style changes in this section-use consistent format throughout

The references have been formatted according to the PLoSOne format.

- Conclusion needs strengthening

The conclusion has been strengthened as recommended and is now: “In conclusion, the present study has clearly shown, at least for the healthy group under investigation, on the one hand that both the experience and the education on the childhood significantly affect the lifelong relationship to the oral cavity. Therefore, parents have to be included into oral hygiene education as early as possible. They serve as role models and children learn many behaviours by simply imitating them. Parents should be educated thoroughly by dentists about oral hygiene measures and should be more closely involved in the process of oral hygiene instruction. Furthermore, negative experiences at a dentist and the resulting anxiety and fear resulted not only in avoidance behaviour regarding visits to the dentist, but also in a poorer attitude towards oral hygiene measures and a decreased toothbrushing motivation. As consequence, early negative experiences at dentists should be avoided best possible. On the other hand, the participants showed a high self-expectation but also high motivation for oral hygiene, which can clearly be judged as a success of the current preventive programmes in Germany. Further research should address a wider population in order to increase generalisability of the results.”

---

## [Decision Letter · Decision Letter 1]

11 Jan 2022

PONE-D-20-29851R1Relationship between dental experiences, oral hygiene education and self-reported oral hygiene behaviourPLOS ONE

Dear Dr. Maxi Katharina Mueller,

Thank you for submitting your manuscript to PLOS ONE. After careful consideration, we feel that it has merit but does not fully meet PLOS ONE’s publication criteria as it currently stands. Therefore, we invite you to submit a revised version of the manuscript that addresses the points raised during the review process.

We look forward to receiving your revised manuscript.

Kind regards,

Tanay Chaubal

Academic Editor

PLOS ONE

Journal Requirements:

Reviewers' comments:

Reviewer's Responses to Questions

**Comments to the Author**

1. If the authors have adequately addressed your comments raised in a previous round of review and you feel that this manuscript is now acceptable for publication, you may indicate that here to bypass the “Comments to the Author” section, enter your conflict of interest statement in the “Confidential to Editor” section, and submit your "Accept" recommendation.

Reviewer #1: (No Response)

Reviewer #3: All comments have been addressed

Reviewer #4: (No Response)

2. Is the manuscript technically sound, and do the data support the conclusions?

Reviewer #1: (No Response)

Reviewer #3: Yes

Reviewer #4: Yes

3. Has the statistical analysis been performed appropriately and rigorously? 

Reviewer #1: (No Response)

Reviewer #3: (No Response)

Reviewer #4: Yes

4. Have the authors made all data underlying the findings in their manuscript fully available?

Reviewer #1: (No Response)

Reviewer #3: Yes

Reviewer #4: Yes

5. Is the manuscript presented in an intelligible fashion and written in standard English?

Reviewer #1: (No Response)

Reviewer #3: Yes

Reviewer #4: Yes

6. Review Comments to the Author

Reviewer #1: Minor comments

The accepted level of significant for p value to be stated.

There was no data presented in table form for S1 Table (S1 Table consists only questions)

The figures in the text (results section) should refer to/cite S4 Table and not S2 Table as there were no figures presented in S2 Table. As such S2 Table to be replaced with S4 Table in the text and followed by respective domains and items.

Reviewer #3: the changes that has been done considering the english in the abstact and introduction is accurate.

Reviewer #4: The authors have presented the manuscript well and all comments have been addressed appropriately. the data given is supporting the conclusion.

7. PLOS authors have the option to publish the peer review history of their article (what does this mean?). If published, this will include your full peer review and any attached files.

Reviewer #1: No

Reviewer #3: No

Reviewer #4: No

---

## [Author Response · Author response to Decision Letter 1]

18 Jan 2022

Dear Dr. Heber, dear reviewers,

Thank you very much for reviewing our manuscript again and for your constructive feedback. We have revised the manuscript in light of the reviewers' comments. The changes are marked in yellow in the manuscript. Please see below for a point-by-point response to the reviewers comments.

Reviewer #1: Minor comments

The accepted level of significant for p value to be stated.

The accepted level of significant for p value is now stated in the section „Materials and Methods – Procedures (Data handling, questionnaire evaluation and statistical analysis) (Line 262-263).

There was no data presented in table form for S1 Table (S1 Table consists only questions)

We have added the results of the questionnaires of the S1 table in tabular form to the "supporting information" under a new S2 Table; the other Supporting Information numbers have been changed accordingly.

The figures in the text (results section) should refer to/cite S4 Table and not S2 Table as there were no figures presented in S2 Table. As such S2 Table to be replaced with S4 Table in the text and followed by respective domains and items.

The references in the results and discussion section have been changed to the respective domain and item numbers as presented in S5 Table (previously S4).

Reviewer #3: the changes that has been done considering the english in the abstact and introduction is accurate.

Thank you for the appreciative comment.

Reviewer #4: The authors have presented the manuscript well and all comments have been addressed appropriately. The data given is supporting the conclusion.

Thank you for the appreciative comment.

---

## [Editor Report · Decision Letter 2]

9 Feb 2022

Relationship between dental experiences, oral hygiene education and self-reported oral hygiene behaviour

PONE-D-20-29851R2

Dear Dr. Maxi Katharina Mueller,

We’re pleased to inform you that your manuscript has been judged scientifically suitable for publication and will be formally accepted for publication once it meets all outstanding technical requirements.

Kind regards,

Tanay Chaubal

Academic Editor

PLOS ONE

---

## [Editor Report · Acceptance letter]

15 Feb 2022

PONE-D-20-29851R2 

Relationship between dental experiences, oral hygiene education and self-reported oral hygiene behaviour 

Dear Dr. Mueller:

I'm pleased to inform you that your manuscript has been deemed suitable for publication in PLOS ONE. Congratulations! Your manuscript is now with our production department. 

Kind regards, 

on behalf of

Dr. Tanay Chaubal 

Academic Editor

PLOS ONE